# Biomarkers of Vascular Injury and Type 2 Diabetes: A Prospective Study, Systematic Review and Meta-Analysis

**DOI:** 10.3390/jcm8122075

**Published:** 2019-11-27

**Authors:** Laura Pletsch-Borba, Cora Watzinger, Renée Turzanski Fortner, Verena Katzke, Lukas Schwingshackl, Solomon A. Sowah, Anika Hüsing, Theron Johnson, Marie-Luise Groß, Sandra González Maldonado, Michael Hoffmeister, Peter Bugert, Rudolf Kaaks, Mirja Grafetstätter, Tilman Kühn

**Affiliations:** 1Division of Cancer Epidemiology, German Cancer Research Center (DKFZ), Im Neuenheimer Feld 581, 69120 Heidelberg, Germany; cora.watzinger@med.uni-muenchen.de (C.W.); r.fortner@dkfz.de (R.T.F.); v.katzke@dkfz.de (V.K.); s.sowah@dkfz.de (S.A.S.); a.huesing@dkfz.de (A.H.); t.johnson@dkfz.de (T.J.); s.gonzalezmaldonado@dkfz.de (S.G.M.); r.kaaks@dkfz.de (R.K.); m.grafetstaetter@dkfz.de (M.G.); t.kuehn@dkfz.de (T.K.); 2Faculty of Medicine, University Medicine Heidelberg, Im Neuenheimer Feld 672, 69120 Heidelberg, Germany; 3Institute for Evidence in Medicine, Medical Center - University of Freiburg, Breisacher Straße 153, 79110 Freiburg, Germany; shwingshackl@cochrane.de; 4Faculty of Medicine, University of Freiburg, Breisacher Straße 153, 79110 Freiburg, Germany; 5Department of Preventive Oncology, National Centre for Tumor Diseases, Im Neuenheimer Feld 460, 69120 Heidelberg, Germany; marie-luise.gross@nct-heidelberg.de; 6Division of Clinical Epidemiology and Aging Research, German Cancer Research Center (DKFZ), Im Neuenheimer Feld 581, 69120 Heidelberg, Germany; m.hoffmeister@dkfz.de; 7Institute of Transfusion Medicine and Immunology, University Medicine Heidelberg, Friedrich-Ebert-Straße 107, 68167 Mannheim, Germany; peter.bugert@medma.uni-heidelberg.de; 8German Red Cross Blood Service Baden-Württemberg-Hessen, Friedrich-Ebert-Straße 107, D-68167 Mannheim, Germany

**Keywords:** Epidemiology, Type 2 Diabetes, E-Selectin, P-Selectin, ICAM3, thrombomodulin, vascular injury biomarkers

## Abstract

Data on biomarkers of vascular injury and type 2 diabetes (T2D) risk from prospective studies are lacking. We evaluated seven biomarkers of vascular injury in relation to T2D. Additionally, a meta-analysis was performed. From the EPIC–Heidelberg cohort, 2224 participants were followed-up from baseline for 16 (median) years. E-Selectin, P-Selectin, intercellular adhesion molecule 3 (ICAM3), thrombomodulin, thrombopoietin, glycoprotein IIb/IIIa and fibrinogen levels were measured in baseline blood samples. The systematic review and meta-analysis included prospective studies identified through MEDLINE and Web of Science that investigated the association between mentioned biomarkers and T2D. The study population included 55% women, median age was 50 years, and 163 developed T2D. ICAM3 was associated with lower T2D risk (fully adjusted HR_highest vs. lowest_ tertile 0.62 (95% CI: 0.43, 0.91)), but no other studies on ICAM3 were identified. Overall, fifteen studies were included in the systematic review and meta-analysis (6,171 cases). E-Selectin was associated with higher T2D risk HR_per SD_: 1.34 (95% CI: 1.16, 1.54; I^2^ = 63%, *n* = 9 studies), while thrombomodulin was associated with lower risk HR_per SD_: 0.82 (95% CI: 0.71, 0.95; I^2^ = 0%, *n* = 2 studies). In the EPIC–Heidelberg, ICAM3 was associated with lower T2D risk. The meta-analysis showed a consistent positive association between E-Selectin and T2D. It was also suggestive of an inverse association between thrombomodulin and T2D, although further studies are needed to corroborate this finding.

## 1. Introduction

An estimated 8% of the world’s adult population live with type 2 diabetes mellitus (T2D), with the global prevalence predicted to increase in the coming years [1]. According to the Global Burden of Disease study, diabetes was the 12th leading cause of death, and total deaths from T2D increased from 2007 to 2017 by nearly 50% [2]. Microvascular dysfunction has been suggested to be one mechanistic pathway linking obesity to increased insulin resistance and, subsequently, T2D [3,4,5,6]. Endothelial dysfunction and insulin resistance are closely related phenomena, as it has been proposed that trans-capillary insulin transport is a rate-limiting step in peripheral insulin action [7]. Endothelial dysfunction can also diminish insulin’s delivery to the interstitium and could thereby limit insulin action [7]. In this context, vascular injury could be a risk factor for T2D. 

The endothelium is a widely distributed organ system and plays an important role in both maintaining blood in its fluid state, as well as in providing controlled haemostasis at sites of vascular injury [8]. Vascular injury can be assessed by circulating concentrations of biomarkers related to inflammation, haemostasis and endothelial activation [4]. Major markers of vascular injury include soluble E-Selectin and P-Selectin, intercellular adhesion molecule 3 (ICAM3), thrombomodulin, thrombopoietin, glycoprotein IIb/IIIa (GP IIb/IIIa) and fibrinogen. E-Selectin is expressed in endothelial cells and is known to be responsible for the neutrophil-rolling interaction with the endothelium [9]. P-Selectin is stored in the Weibel–Palade bodies of the alpha-granules of platelets and is expressed in endothelial cells [10,11,12]. ICAM3 may mediate interactions between leucocytes and endothelial cells in atherosclerosis, becoming functionally altered during apoptosis to mediate clearance of apoptotic leucocytes through suggested generation of “eat me” signals [13,14,15]. Thrombomodulin is a membrane glycoprotein expressed by endothelial cells of all vessels (arteries, veins, capillaries and lymphatics) that converts thrombin from a procoagulant to an anticoagulant enzyme thus modulating secondary haemostasis [16,17]. Thrombopoietin is a humoral growth factor produced by the liver and kidney and is the most important growth factor in the regulation of megakaryocytes development and platelet synthesis [18,19,20,21]. GP IIb/IIIa is a receptor expressed on the surface of activated platelets, essential for platelet aggregation via binding to fibrinogen [22].

We have previously evaluated the mentioned biomarkers in relation to cardiovascular risk factors and myocardial infarction (MI) risk in a population-based study. While we did not observe prospective associations with MI, there were several associations with cardiovascular risk factors [23]. For example, plasma levels of P-Selectin, E-Selectin and ICAM3 were higher among study participants with prevalent diabetes. Moreover, thrombomodulin has been shown to be inversely associated with T2D in the population-based MONICA/KORA study [24], and further previous studies indicate that E-Selectin levels may be associated with increased diabetes risk [4,5,25,26,27,28,29,30]. Considering the biological properties of the mentioned biomarkers, our own analyses on cardiovascular risk factors, and previous findings from other population-based studies, we decided to evaluate the aforementioned biomarkers in relation to T2D risk in the population-based EPIC–Heidelberg study. Hypothesizing that vascular injury, measured through circulating biomarkers, could be a risk factor for T2D, we studied the associations between the biomarkers of vascular injury and incidence of T2D in a subcohort EPIC–Heidelberg including women and men aged 35–65 at baseline, who were followed-up for 16 years. We also performed a systematic review on the evidence for associations between these markers and incident T2D, and a meta-analysis in order facilitate the contextualization of our results.

## 2. Materials and Methods

### 2.1. Study Population (Original Research)

The EPIC–Heidelberg cohort included 25,540 participants aged 35–65 years, who were recruited between 1994 and 1998 from the general population of Heidelberg, Germany and the surrounding areas [31,32]. At baseline, participants completed detailed questionnaire-based and interviewer-administered assessments on diet, lifestyle factors, medication use and health status, and anthropometric measurements were taken. Blood samples were collected and processed using a standardized protocol, and have been stored in gas phase liquid nitrogen at −150° Celsius. Details of blood sample collection and processing have been previously described in detail [32]. The present analyses on plasma biomarkers and T2D risk were conducted using a randomly selected subcohort of approximately 10% of the original cohort. This subcohort was initially drawn for case-cohort analyses on biomarkers in relation to risks of common cancers and cardiovascular diseases [33,34] and included 2224 EPIC–Heidelberg participants free of diabetes at baseline (*n* = 194 prevalent cases excluded). Follow-up was conducted through 31st December 2012. The Ethics Committee of the Heidelberg University Hospital approved the study and all participants gave written informed consent.

### 2.2. Ascertainment of Incident Diabetes Mellitus (Original Research)

Participants have been followed through a combination of active and passive follow-up methods since baseline [31], and completed follow-up questionnaires at 2–3 year intervals following recruitment, with response rates of circa 95% [35]. Incident cases of T2D were identified based on self-report of a new diagnosis of diabetes, use of diabetes relevant medication, or change in diet due to disease. Self-reported cases were validated by a trained study physician using diagnostic records obtained from the treating physician. In addition to diagnoses reported on questionnaires, information from death certificates and record linkage with the major hospital in the area were conducted, again followed by a validation based on individual records. 

### 2.3. Laboratory Methods (Original Research)

Details on the laboratory methods have been previously described [23]. Briefly, E-Selectin, P-Selectin, ICAM3 and thrombomodulin were measured with the “Quiplex SQ 120” instrument from Meso Scale Discoveries (MSD Maryland, USA) using the “human vascular injury kit I” multiplex assay kit. Thrombopoietin was measured by electrochemoluminescence immunoassays (ECLIA) (“U-Plex TPO Assay” kit from MSD). Enzyme-linked immunosorbent assays (ELISA) was used to measure GP IIb/IIIa and fibrinogen levels, using the essay kits “ab108851” from Abcam (Cambridge, UK) and “KA0475” from Abnova (Heidelberg, Germany). Within- and between-batch coefficients of variation were as follows (within-batch CV (between-batch CV)): P-Selectin 3.3% (9.1%), E-Selectin 3.6% (10.6%), thrombomodulin 3.8% (10.1%), thrombopoietin 4.6% (19.5%), GP IIb/IIIa 5.5% (46.9%) and ICAM3 7.5% (10.2%) (Appendix A) [33]. All biomarkers showed good biological reproducibility in a pilot study carried out prior to the present analyses including 78 participants from the EPIC–Heidelberg subcohort, with Spearman’s coefficients (ρ) for intra-individual correlations over one year of 0.88 (E-Selectin), 0.80 (P-Selectin), 0.69 (ICAM3), 0.63 (thrombomodulin), 0.73 (thrombopoietin), and 0.51 (GP IIb/IIIa) (Appendix A) [23,36]. Biological reproducibility of fibrinogen was not assessed, as exhaustive previous repeatability analyses indicated low intra-individual variation over time [37]. The batch mean-centering method was used for batch standardization [38]. Inter-correlations among the measured biomarkers were mild (ρ < 0.4), except for E-Selectin and P-Selectin (ρ = 0.56) [23]. 

### 2.4. Covariates Assessment

Covariates assessment is described in detail in the Appendix A.

### 2.5. Systematic Review and Meta-Analysis

We searched MEDLINE and Web of Science databases following PRISMA guidelines [39] (through 4th June 2019), without language restriction, using the search strategies provided in the Appendix A. Titles, abstracts and full-texts were screened independently by 2 reviewers in the first step (LPB and CW). As double check, 2 reviewers have each re-screened half of the total list of titles (SAS and TK). In case of disagreement between the reviewers, consensus was reached through discussion. Inclusion criteria: (i) Population-based prospective cohort, case-cohort and nested case-control studies; (ii) that evaluated circulating concentrations of at least one of the 7 aforementioned vascular dysfunction biomarkers; (iii) in participants free of T2D at baseline. We examined the reference lists of the selected papers to find additional relevant articles. In cases where potentially relevant results were referred to as “unpublished data” in reviewed full-text articles, we contacted the study authors. The following data was extracted from the full texts: study design, baseline characteristics of the study population, number of participants without T2D at baseline, incident cases of T2D, mean follow-up, definition of T2D, crude and adjusted odds ratios, relative risks or HRs with respective 95% confidence intervals (CI) and adjusting covariates included in the multivariate analyses (Tables 3 and 4). Quality of the studies was assessed using the adapted Newcastle-Ottawa Scale (Appendix A) [40]. International Prospective Register of Systematic Reviews PROSPERO 2019: CRD42019125922.

### 2.6. Statistical Analyses 

Continuous variables are described as median (percentile25; percentile75) and categorical variables as numbers (percentages). T-Tests (for continuous variables) and Chi-square (for categorical variables) tests were used for calculation of *p*-values for differences in covariates between T2D cases and non-cases. Values of all plasma biomarkers were log10 transformed so that the distribution better approximated a normal distribution, and batch-standardized (mean of 0 and standard deviation SD of 1). We categorized biomarker levels by tertiles using thresholds based on sex-specific distributions in the cohort, as biomarker levels were generally higher in samples of men compared to those of women. Percentage of missing values in the covariates was 8% (Appendix A). Missing values were multiply imputed under the missing at random assumption, using a fully conditional algorithm, with number of imputations set to 5; results from the 5 imputed datasets were pooled using SAS procedure PROC MIANALYZE.

We used Cox proportional hazards regression models to calculate hazard ratios (HRs) and 95% confidence intervals (CI) [41]. Each plasma biomarker was analysed as an individual risk factor for incident T2D. We used age as a time scale, and all participants were left-truncated at the time of blood sampling, and censored at T2D diagnosis, loss to follow-up, or end of the study follow-up, whichever occurred first. Model 1 was adjusted for age and sex. Model 2 was additionally adjusted for body mass index (BMI) (kg/m^2^), alcohol consumption (g/day in the past year), smoking status (never, past quit ≥10 years ago, past quit <10 years ago, current <15 cigarettes/day, current ≥15 cigarettes/day), physical activity (Cambridge index, inactive/moderately inactive vs. moderately active/active), education level (primary, secondary and university), self-reported hypertension (yes/no), glycated haemoglobin (HbA1c) and C-Reactive protein (CRP). All covariates were determined a priori. Further adjustments for fibre intake, meat consumption, and coffee intake were additionally performed in sensitivity analyses. Extended correlation tests based on Schoenfield residuals indicated no violations of the proportional hazards assumption. Potential effect modifications of sex and age were tested including a multiplicative term into the Cox regression models. Non-linear associations between the biomarkers and T2D risk were tested using restricted natural splines, and the best fitting regression model was chosen based on the lowest Akaike Information Criteria. Additionally, we estimated to which extent the vascular injury biomarkers improve the prediction of absolute T2D risk beyond the covariates mentioned above using the concordance (C-) statistic, also known as area under the ROC curve (AUROC), calculated by logistic regression analyses. Our basic model for risk prediction included all the variables used as covariates from multivariable Cox regression analyses. Biomarkers were then added to the basic model to evaluate changes in the AUROC. The improvement in discriminative capacity in terms of C-statistics between nested models with and without biomarkers was tested by a likelihood-ratio test [42].

For the meta-analysis, in order to compare effect estimates between studies that analysed biomarker concentrations in quantiles and those that analysed concentrations as continuous variables, we converted effect estimates from the quantile-based analyses into expected estimates if the biomarker had been evaluated as a continuous variable (per SD increase in biomarker concentration); details on the statistical methods and assumptions are included in the Appendix A. If only sex-specific effect estimates were reported, those results were pooled using fixed effects models. HRs were calculated using random-effects meta-analysis as proposed by Hedges and Vevea [43]. When multiple multivariable models were included in a publication, we included the effect estimate adjusted for the maximum number of risk factors, but without further adjustment for other biomarkers of vascular injury (if available). If such a model was not available, sensitivity analyses were performed excluding the respective study. Sensitivity analyses by subgroups of study characteristics such as location, fasting vs. non-fasting blood collection and adjustment for covariates were performed (Appendix A). Heterogeneity between studies was measured by the I^2^ statistic [44]. Publication bias was not assessed due to the small number of included studies [45]. A two-sided p-value of less than 0.05 denoted statistical significance. SAS version 9.4 (SAS Institute, Cary, North Carolina, USA) for Windows was used to conduct descriptive analyses, imputation of missing data, and Cox regression analyses, while R statistical software version 3.4.3 was used to conduct the meta-analyses (Metafor R package), to test for non-linear associations in the Cox regression analyses, and to produce graphs.

## 3. Results

### 3.1. Biomarkers of Vascular Injury and Type 2 Diabetes Risk in the EPIC–Heidelberg

Among 2224 EPIC–Heidelberg participants, 55% (*n* = 1217) were women, 163 participants (7.3%) developed T2D over a median follow-up of 16 years (p25; p75: 15; 17 years). Participants who developed T2D were more frequently men, had older age, higher prevalence of hypertension, higher BMI, lower educational level and lower levels of physical activity at baseline. Further, these participants had lower high-density lipoprotein (HDL) levels and higher levels of CRP, low-density lipoprotein (LDL), triglycerides, and HbA1c, as well as E-Selectin, P-Selectin, thrombomodulin, GP IIb/IIIa and fibrinogen (Table 1). Median biomarker concentrations for each tertile among women and men are described in Appendix A.

Table 2 shows associations between biomarkers of vascular injury and T2D risk. In fully adjusted models, E-Selectin was associated with increased T2D risk among study participants in the second, but not the third tertile of E-Selectin concentrations, with HRs of 2.13 (95%CI 1.38, 3.29) and 1.44 (0.93, 2.22), respectively. ICAM3 was statistically significantly associated with decreased risk of T2D (HR_highest vs. lowest tertile_: 0.62 (95%CI 0.43, 0.91)). With respect to thrombomodulin, there were no significant differences in diabetes risk across tertiles, despite a non-significant linear trend for an inverse association (*p* = 0.06). When analysing linear trends using continuous values of biomarkers, none were statistically significantly associated with T2D risk. Further adjustments for fibre intake, meat consumption, and coffee intake did not change the results considerably. No statistically significant interactions between any of the biomarkers with age or sex in relation to diabetes risk were observed. There was evidence for non-linearity in the association between E-Selectin levels and T2D risk, as the model which best fitted the data included natural splines with two knots (Appendix A), while none of the other biomarkers showed non-linear associations with T2D risk.

Diabetes risk prediction analyses showed an AUC of 0.840 (95%CI 0.809, 0.870) for the basic model (i) (without any vascular injury biomarker). There was statistically significant improvement in the prediction upon addition of (ii) E-Selectin, ICAM3 and thrombomodulin, AUC 0.842 (95%CI 0.811, 0.874), *p* = 0.02. When (iii) adding all seven biomarkers to the model, AUC 0.844 (95%CI 0.812, 0.875), *p* = 0.15, there was no significant increase in the prediction ability, in comparison to the basic model (i). 

### 3.2. Systematic Review and Meta-Analysis

We identified 2720 unique articles in our initial database searches (Figure 1). After title, abstract and full-text screening, 22 eligible articles that analysed the associations between vascular injury biomarkers and risk of T2D in population-based studies were identified. No relevant abstracts without full-text were found. One article was excluded because despite data availability, the association between fibrinogen and T2D risk was not assessed [46], two articles did not analyse any of the biomarkers of interest [47,48], one article did not report the single effect of fibrinogen, but only its effect combined with CRP [49], and individual findings from the MONICA/KORA [6,50], ARIC [51,52], and Women Health’s Initiative [27,53] studies were each reported in two publications, while another study reported change in fibrinogen, and not baseline fibrinogen concentration, in relation to T2D [54]. In case of repeated publications, the one with the higher case number and/or more complete model adjustment was included. Overall 15 studies, including the present one, were included in the systematic review and meta-analysis (Table 3 and Table 4). The majority of the studies included populations from the USA or Europe who were middle-aged (mean age range: 32; 64), and follow-up durations ranged from 4 to 16 years. Table 3 and Table 4 summarize studies and participants characteristics. All included articles were published in English and after 1999. Studies were of high quality, ranging from 7 to 9 out of 9 stars (Newcastle-Ottawa Scale, Table 4, Appendix A). We selected four nested case-control studies [5,25,27,28], seven cohort studies [4,29,52,55,56,57,58], one subset of a cohort study [30], and two case-cohort studies [6,24]. 

Eight studies investigated the relationship between E-Selectin and type 2 diabetes risk [4,5,6,25,27,28,29,30]. All studies reported higher E-Selectin concentrations to be associated with higher T2D risk, with continuous multivariable-adjusted effect estimates ranging from 1.18 (95%CI 0.90,1.55) [29] to 1.58 (95%CI 1.45, 1.73) [27]. All three studies [4,30,56] that investigated the association between P-Selectin and type 2 diabetes risk found no significant relationships. In the single prior study on thrombomodulin, a borderline inverse association (0.79 (95%CI 0.62, 1.00) per SD increase in concentration, *p* = 0.047) was reported [24]. In the 5 studies on fibrinogen and T2D, effect estimates ranged from 0.91 (95%CI 0.75, 1.11) [56] to 7.18 (95%CI 1.63, 31.53) [58] in the continuous analyses, and from 1.0 (95%CI 0.8, 1.3) [57] to 1.2 (95%CI 1.0, 1.5) [52] when comparing highest to lowest fibrinogen quartiles. No studies on ICAM3, thrombopoietin, and GP IIb/IIIa in relation to type 2 diabetes risk were found. 

### 3.3. Meta-Analysis

E-Selectin was associated with higher T2D risk in the meta-analysis (HR_per SD_: 1.34 (95% CI: 1.16, 1.54); I^2^ = 63%; *n* = 9 studies) (Figure 2). Analyses using the less adjusted models showed similar results (exclusion of one study that did not report age-and –sex adjusted models [30]) (HR_per SD_: 1.45 (95% CI: 1.25, 1.70); I^2^ = 78%; *n* = 8 studies). As one study on E-Selectin [4] did not report a multivariable model only adjusted for epidemiological covariates (only also adjusted for other biomarkers), we performed sensitivity analyses excluding this study. Results changed only marginally (HR_per SD_ 1.32 (1.14, 1.54), I^2^ = 67%, Appendix A). Also, exclusion of the present study only led to marginal changes (Appendix A). When including in the meta-analysis only those reported estimates from models with the most comprehensive adjustment (i.e., with additional biomarkers beyond epidemiological covariates) from each study, results changed only slightly. At the same time, there was a tendency for weaker associations in studies with an overall higher degree of adjustment compared to those with less comprehensive adjustment (Appendix A). Sensitivity analyses on E-Selectin selecting Herder et al. [50] instead of Thorand et al. [6] (same study population) showed no material differences in the results (Appendix A). Similarly, no material changes were observed when using two estimates for men and women from Thorand et al. [6], instead of pooling them before entering in the meta-analysis (Appendix A). The association between thrombomodulin and type 2 diabetes risk was assessed in one other study [24] besides ours. A significant inverse association was observed between thrombomodulin and T2D risk in the meta-analysis (HR_per SD_: 0.82 (0.71, 0.95), I^2^ = 0%). Although the previous study reported non-transformed estimates, and we used log-10 transformed estimates, this transformation changed our results only marginally. No statistically significant association was observed between P-Selectin (HR_per SD_ 0.98 (0.89, 1.07); I^2^ = 0%, *n* = 4 studies) or fibrinogen and T2D risk (HR_per SD_: 1.01 (95%CI 0.91, 1.13); I^2^ = 52%, *n* = 6 studies) in the meta-analysis (Figure 2). 

## 4. Discussion

In the present analyses of data from the EPIC–Heidelberg, E-Selectin showed a non-linear association with T2D risk, while ICAM3 showed an inverse association with T2D over a median 16 year follow-up period. In the meta-analysis, higher levels of E-Selectin were associated with increased T2D risk, while increased thrombomodulin levels were related to lower T2D risk. The other selected biomarkers showed no association with T2D. No previous studies that explored the association between ICAM3 and T2D were found.

The observation of a positive association between E-Selectin and diabetes risk in our meta-analysis is in line with mechanistic data indicating that E-Selectin, which is expressed on the surface of activated endothelial cells and binds to ligands present on leucocytes, promotes endothelial activation and injury [60]. In a previous cross-sectional study in the EPIC–Heidelberg population, we observed that E-selectin levels are associated with higher BMI, alcohol consumption, LDL-cholesterol, total-cholesterol, and with lower education levels as well as lower HDL-cholesterol [23]. Hence, higher E-Selectin levels may reflect a more unfavourable lifestyle pattern and lower socio-economic status, and co-occur with other cardio-metabolic alterations that lead to T2D. Similarly, whereas in subgroup analyses the positive association between E-Selectin and T2D was present in almost all strata, it was slightly weaker in studies with adjustment for CRP, hypertension, and glucose. Yet, these weaker pooled associations in studies with more comprehensive adjustment should be interpreted carefully, given the lower number of studies. 

Our analyses suggested a hat-shaped non-linear association between E-Selectin and T2D in the EPIC–Heidelberg Study. In contrast, a previous meta-analysis by Qiu et al. suggested a non-linear j-shaped association [61], although this association was not statistically significant, and there is no strong indication for non-linearity from prior single studies. Therefore, and due to the lack of an obvious biological explanation for a potential non-linear relationship between E-Selectin levels and T2D risk, we cannot rule out that the non-linear hat-shaped association based on 163 incident cases in our study was due to chance.

We observed an 18% lower risk of T2D per 1-SD thrombomodulin in our meta-analysis. Besides our study, only one other study investigated thrombomodulin in relation to T2D, and both included only participants from southern Germany [24]. The association observed in the previous study (HR 0.79 (0.62, 1.00), *p* = 0.047) was very similar to the observed in the current one (HR 0.84 (0.71, 1.00), *p* = 0.06), although in the EPIC–Heidelberg it was not statistically significant. The fact that in the MONICA/KORA study a higher number of incident T2D cases occurred may explain this slight difference. Overall, the findings from both studies may support the notion of thrombomodulin exerting anti-coagulative effects [62], and may in turn indicate that vascular injury is implicated in early phases of T2D development, rather than only constituting a late pathophysiological consequence of established T2D. However, further, larger studies are needed to corroborate the present meta-analysis result based on only two cohorts.

The inverse association found in the EPIC–Heidelberg cohort between ICAM3 and T2D risk has not yet been reported from any previous study. ICAM3, a glycoprotein shed by activated leukocytes [15], was demonstrated to be responsible for tethering apoptotic cells to phagocytes, and thus, indicates an increased inflammatory state [14]. Furthermore, ICAM3 has been shown to promote carcinogenesis via pro-inflammatory effects in experimental studies [63]. Results from the present study were unexpected considering that ICAM3 was previously positively associated with prevalent T2D in our cohort, in cross-sectional analyses [23]. Additional studies on this marker are required, and we cannot rule out that the reported association was due to chance, also considering that no significant association between ICAM3 on the continuous scale and T2D was observed.

Three studies [4,30,56], besides the present one, have analysed the association between P-Selectin and T2D risk; all reported null findings. We found no prior studies that examined the association between thrombopoietin and GP IIb/IIIa in relation to T2D risk, and we observed no statistically significant associations between these biomarkers and T2D risk in the current study. Fibrinogen and T2D has been investigated in five prior studies [52,55,56,57,58], with all but one [58] (in which fibrinogen was positively associated with diabetes risk), reporting no associations. In our study, a positive association was strongly attenuated and no longer statistically significant upon adjustment for confounders. The single positive association reported from the Strong Heart Study may be driven by different ethnicity (American Indians vs. non-Hispanic white from other studies), the younger age of the study population, or the different range of covariates used for statistical adjustment (e.g., body fat instead of BMI) [58].

A potential improvement of T2D risk prediction models upon addition of E-Selectin has been reported by the MONICA/KORA study [50]. The present study suggests further that besides E-Selectin, the inclusion of thrombomodulin and ICAM3 may slightly increase the AUC of prediction models, although further prospective studies on these biomarkers are needed. Moreover, associations between E-Selectin and T2D risk showed heterogeneity across study populations, which may speak against its usefulness for risk stratification. The observation of similar inverse associations with T2D in the two available studies on thrombomodulin (EPIC–Heidelberg, MONICA/KORA) may be related to the fact that both study populations comprised white adults from southern Germany with similar characteristics, and further studies on thrombomodulin and T2D in populations with different ethnic and socio-economic characteristics are needed. In general, it should be noted that the present study and meta-analysis did not have the purpose to generate or validate a T2D prediction model, and that multibiomarker studies with external validation are required to assess whether E-Selectin, ICAM3, and thrombomodulin are potential predictors of absolute T2D risk.

Strengths of our study include the comprehensive set of covariates available for statistical adjustment, including HbA1_c_ and CRP. Furthermore, we previously demonstrated that the biomarkers investigated in this study had good one-year reproducibility (Spearman correlations ranges 0.51 to 0.88) [36]. Besides that, we conducted a meta-analysis, in additional to our original research, which synthesizes available data on this topic. Finally, our results on E- and P-Selectin were comparable to the ones reported by a meta-analysis that was performed in parallel to this one [61]. Interestingly, risk estimates for associations between E-Selectin and T2D risk were much stronger in magnitude in this meta-analysis by Qiu et al. [61], which may be due to the fact that E-Selectin concentrations were modelled on the µg/mL scale. For our analyses, we decided to use the ng/mL scale, as E-Selectin concentrations were below 0.1 µg/mL in the majority of blood samples of our study population and others. Thus, risk estimates for E-Selectin in relation to T2D risk from our meta-analysis are lower, but still consistent with those reported by Qiu et al. [61]. 

Our study also has limitations that must be acknowledged. Although we had a relatively long period of follow-up, we only observed 163 incident cases of T2D in this subcohort of the EPIC–Heidelberg, with limited power for subgroup analyses. The selection of biomarkers for the meta-analysis was opportunistic in that we only included biomarkers which were measured in our own cohort. The relatively low case number may be due to good general health status of the EPIC–Heidelberg participants at baseline. Although this potential “healthy cohort effect” may also have affected our results (towards the null), our results are in agreement with previous studies. Additionally, results from the EPIC–Heidelberg cannot be generalized to non-white populations, and the studies included in the review and meta-analysis were also performed in predominantly white populations, which highlight the need for multi-ethnic studies. Further, these results may not be generalizable to high-risk populations, such as obese individuals. In addition, there may be other interesting biomarkers of vascular injury such as ICAM1 [61] that were not covered by the present meta-analysis, as we focused on biomarkers that had been measured in EPIC–Heidelberg. Associations of vascular injury biomarkers with complications of T2D were out of the scope of this review, and studies on these biomarkers in the context of diabetes complications are needed. Lastly, as in all observational studies, we cannot exclude that the associations we found may be affected by residual confounding, even though we have statistically controlled for a wide range of T2D risk factors. 

## 5. Conclusions

The present findings reiterate that increased E-Selectin, a marker of vascular injury, is associated with increased T2D risk, while higher circulating thrombomodulin levels, indicative of vessel protection, may be related to a decreased risk of developing T2D. Thus, endothelial dysfunction seems to be implicated in early phases of T2D development many years before diagnosis. While E-Selectin and thrombomodulin may be interesting candidate biomarkers for diabetes risk prediction models, further multi-biomarker studies are needed to assess and validate their predictive capacity in a targeted manner.

## Figures and Tables

**Figure 1 jcm-08-02075-f001:**
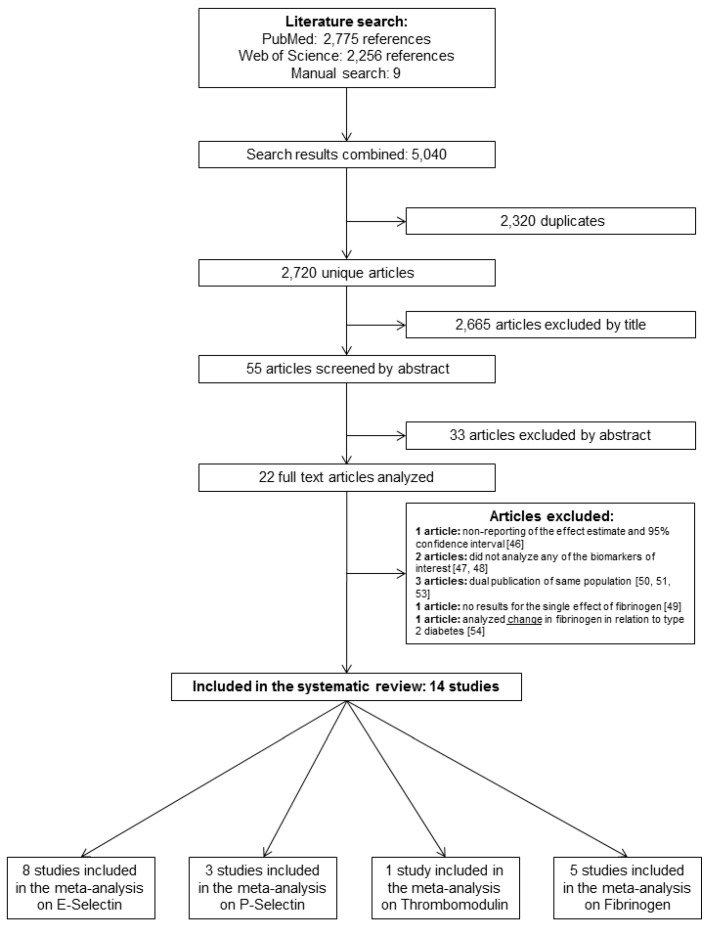
Flowchart illustrating selection of studies for the systematic review and meta-analysis. The present study using EPIC–Heidelberg data is not included.

**Figure 2 jcm-08-02075-f002:**
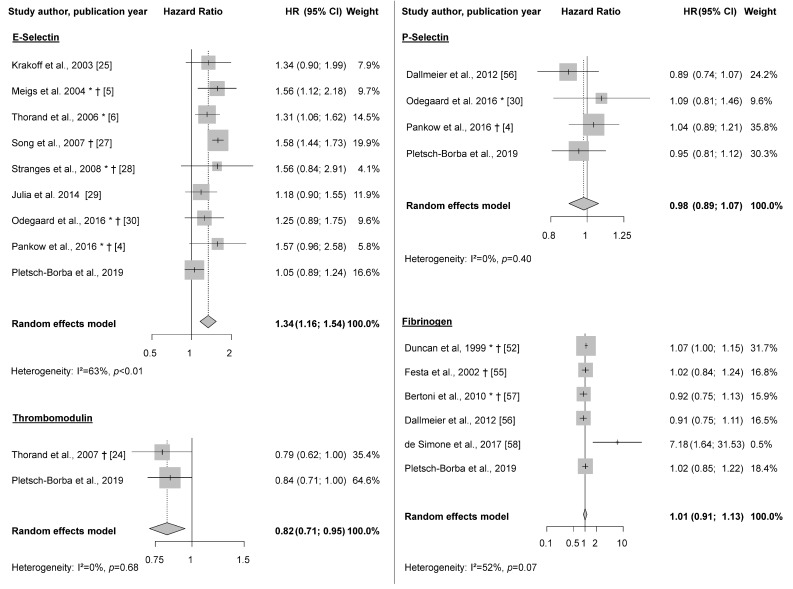
Meta-analysis on biomarkers of vascular injury and type 2 diabetes risk. Random effects meta-analysis on E-Selectin, P-Selectin, Thrombomodulin, and Fibrinogen and type 2 diabetes risk. All effect estimates and confidence intervals derived from the multivariable-adjusted models, as described in Table 4. * Data derived from transformation of quantiles analyses into “per SD”, † No log-transformation of the original circulating biomarker concentration performed, except for standardization (mean = 0, SD = 1).

**Table 1 jcm-08-02075-t001:** Baseline EPIC–Heidelberg subcohort characteristics (*n* = 2224).

	Type 2 Diabetes Cases (*n* = 163)	Non-Cases (*n* = 2061)	*p*-Value
Age at recruitment (years) *	54.2 (47.4; 58.7)	50.34 (43.0; 57.2)	0.09
Women †	62 (38.0%)	1155 (56.0%)	<0.001
Hypertension (yes) †	94 (57.7%)	700 (34.0%)	<0.001
BMI (kg/m^2^) *	29.2 (27.6; 31.8)	24.9 (22.5; 27.6)	0.05
Height (cm) *	170.2 (163.6; 176.0)	169.0 (162.8; 175.7)	0.25
Weight (kg) *	85.3 (77.0; 94.0)	72.2 (63.0; 82.0)	0.33
Waist circumference (cm) *	99.6 (92.7; 106.0)	86.0 (76.0; 94.5)	0.10
Alcohol intake at baseline (g/day) *	11.1 (3.1; 24.5)	10.3 (2.9; 24.4)	<0.01
Education level †			<0.001
Primary School †	67 (41.1%)	526 (25.5%)	
Secondary School †	65 (39.9%)	851 (41.3%)	
University Degree †	31 (19.0%)	684 (33.2%)	
Smoking Status †			0.35
Never †	68 (41.7%)	906 (44.0%)	
Former, quit <10yrs ago †	29 (17.8%)	456 (22.1%)	
Former, quit ≥ 10yrs ago †	23 (14.1%)	226 (11.0%)	
Current <15 cigarettes/day †	21 (12.9%)	261 (12.7%)	
Current ≥ 15 cigarettes/day †	22 (13.5%)	207 (10.0%)	
Aspirin use (yes) †	4 (2.5%)	57 (2.8%)	0.81
Antithrombotic drug use (yes) †	1 (0.6%)	18 (0.9%)	0.72
Physical Activity ^b^ (Cambridge index) †			0.03
Inactive/moderately inactive †	89 (54.6%)	946 (45.9%)	
Moderately active/active †	74 (45.4%)	1115 (54.1%)	
CRP (mg/l) *	2.0 (0.9; 3.6)	0.91 (0,47; 2,29)	0.03
LDL (mmol/l) *	4.2 (3.7; 4.9)	3.9 (3,3; 4,6)	0.90
Triglycerides (mmol/l) *	2.2 (1.6; 3.4)	1.5 (1.0; 2.2)	<0.001
HDL (mmol/l) *	1.2 (1.0; 1.5)	1.4 (1.2; 1.8)	<0.01
Total Cholesterol (mmol/l) *	5.9 (5.4; 6.9)	5.8 (5.1; 6.5)	0.72
HbA_1c_ (mmol/mol, %) *	38.0 (35.0; 41.0), 5.6% (5.4%; 5.9%)	34.0 (32.0; 36.0), 5.3% (5.1%; 5.4%)	<0.001
E-Selectin (ng/mL) *	11.4 (8.9; 14.9)	9.6 (6.9; 13.2)	0.32
P-Selectin (ng/mL) *	29.6 (24.0; 38.0)	27.2 (21.5; 33.9)	0.08
ICAM3 (ng/mL) *	0.42 (0.36; 0.55)	0.44 (0.36; 0.55)	<0.01
Thrombomodulin (ng/mL) *	3.0 (2.6; 3.5)	2.9 (2.4; 3.38)	0.32
Thrombopoietin (pg/mL) *	334.2 (294.4; 398.4)	342.8 (287.6; 409.8)	<0.001
Glycoprotein IIb/IIIa (ng/mL) *	402.9 (324.8; 525.6)	382.1 (313.4; 490.7)	0.84
Fibrinogen (µg/mL) *	3,998 (3643; 4471)	3,731 (3350; 4204)	0.32

Data presented as *median (p25; p75) for continuous variables, and †n (%) for categorical variables. Abbreviations: CRP C-reactive protein, LDL low-density lipoprotein, HDL high-density lipoprotein, HbA_1c_ glycated haemoglobin, and ICAM3 intercellular adhesion molecule 3. *p*-values for the differences between means (continuous) and frequencies (categorical) of described parameters between participants who developed T2D and who did not develop T2D. b Physical activity measured through the Cambridge index and categorized into inactive/moderately inactive vs. moderately active/active.

**Table 2 jcm-08-02075-t002:** Associations between biomarkers of vascular risk and type 2 diabetes risk in the EPIC–Heidelberg.

		Model 1		Model 2	
	Case Number	HR (95%CI)	*p*-Value	HR (95%CI)	*p*-Value
E-Selectin (ng/mL)					
Tertile 1	32	Ref.		Ref.	
Tertile 2	63	2.05 (1.34, 3.14)	<0.01	2.13 (1.38, 3.29)	<0.01
Tertile 3	67	2.14 (1.40, 3.26)	<0.001	1.44 (0.93, 2.22)	0.10
Continuous (per SD)	162	1.39 (1.17, 1.64)	<0.001	1.05 (0.90, 1.24)	0.54
P-Selectin (ng/mL)					
Tertile 1	46	Ref.		Ref.	
Tertile 2	51	1.08 (0.72, 1.61)	0.71	0.82 (0.55, 1.24)	0.36
Tertile 3	65	1.37 (0.94, 2.00)	0.10	0.81 (0.54, 1.21)	0.30
Continuous (per SD)	162	1.20 (1.02, 1.42)	0.03	0.95 (0.81, 1.12)	0.57
ICAM3 (ng/mL)					
Tertile 1	66	Ref.		Ref.	
Tertile 2	46	0.67 (0.46, 0.98)	0.04	0.66 (0.45, 0.98)	0.04
Tertile 3	50	0.74 (0.51, 1.07)	0.11	0.62 (0.43, 0.91)	0.01
Continuous (per SD)	162	0.92 (0.79, 1.08)	0.32	0.89 (0.76, 1.03)	0.13
Thrombomodulin (ng/mL)					
Tertile 1	49	Ref.		Ref.	
Tertile 2	54	1.07 (0.73, 1.58)	0.73	0.93 (0.63, 1.39)	0.74
Tertile 3	59	1.12 (0.76, 1.64)	0.56	0.94 (0.63, 1.40)	0.74
Continuous (per SD)	162	0.97 (0.82, 1.14)	0.69	0.84 (0.71, 1.00)	0.06
Thrombopoietin (pg/mL)					
Tertile 1	54	Ref.		Ref.	
Tertile 2	60	1.13 (0.78, 1.63)	0.51	1.06 (0.72, 1.54)	0.78
Tertile 3	49	0.86 (0.59, 1.27)	0.45	0.92 (0.61, 1.37)	0.67
Continuous (per SD)	163	0.91 (0.78, 1.07)	0.26	0.95 (0.81, 1.12)	0.56
Glycoprotein IIb/IIIa (ng/mL)					
Tertile 1	55	Ref.		Ref.	
Tertile 2	51	0.93 (0.63, 1.36)	0.70	0.79 (0.53, 1.18)	0.25
Tertile 3	56	1.02 (0.70, 1.47)	0.94	0.86 (0.59, 1.26)	0.43
Continuous (per SD)	162	1.01 (0.87, 1.18)	0.85	0.95 (0.82, 1.10)	0.51
Fibrinogen(µg/mL)					
Tertile 1	33	Ref.		Ref.	
Tertile 2	45	1.31 (0.84, 2.06)	0.23	0.96 (0.60, 1.52)	0.86
Tertile 3	85	2.46 (1.63, 3.69)	<0.001	1.05 (0.67, 1.64)	0.82
Continuous (per SD)	163	1.46 (1.25, 1.71)	<0.001	1.02 (0.85, 1.22)	0.82

Model 1 adjusted for age and sex. Model 2 additionally adjusted for height (m), waist circumference (cm), alcohol consumption (g/day in the past year), smoking status (never, past quit ≥10 years ago, past quit <10 years ago, current <15 cigarettes/day, current ≥15 cigarettes/day), physical activity (Cambridge index), education level (primary, secondary and university), hypertension (yes/no) and glycated haemoglobin (HbA_1c_). Continuous variables were log10-standardized (mean = 0, SD = 1 and batch standardized), p-trend indicates p-value for the quantitative (log10-linear, standardized) effect; HR indicates hazard ratio, CI confidence interval, and ICAM3 intercellular adhesion molecule 3.

**Table 3 jcm-08-02075-t003:** Characteristics of the studies included in the systematic review on vascular injury biomarkers and type 2 diabetes risk.

First Author, Year	Study	Ethnicity (%)	Age (Years, Mean)		BMI (kg/m^2^, Mean)		Hypertension (% Yes)		Current Smokers (% Yes)		Participants without Type 2 Diabetes at Baseline	Incident Type 2 Diabetes Cases	Mean Follow-Up (Years)	Definition of Type 2 Diabetes Diagnosis
			Developed Type 2 Diabetes											
			No	Yes	No	Yes	No	Yes	No	Yes				
Duncan, 1999 [52]	ARIC	W: 78, AA: 21	54 among all (median)		26 among all (median)		30 among all		25 among all		12,330	1335	7	Reported physician diagnosis, fasting plasma glucose ≥7 mmol/L, casual glucose ≥11.1 mmol/L, or antidiabetic medication use
Festa, 2002 [55]	Insulin Resistance Atherosclerosis Study	Cases W: 37, B: 28, H: 35 Non-cases W: 41, B: 26, H:33	55	56	28	31	30	45	NA	NA	1,047	144	5.2	WHO criteria [59]
Krakoff, 2003 [25]	Longitudinal health study in Pima Indians	Pima Indians	32	33	36	36	NA	NA	NA	NA	142	71	4.6(cases) 6.8(controls)	WHO criteria [59]
Meigs, 2004 [5]	Nurses’ Health Study	W: 95, Non-white: 5	56	56	26	30	NA	NA	13	14	1,446	700	10	Treatment with either insulin or an oral hypoglycaemic agent, at least 1 classic symptom of diabetes plus an elevated plasma glucose level, or an elevated plasma glucose level on 2 occasions. Elevated plasma glucose was defined as at least 140 mg/dL (≥7.8 mmol/L) fasting, at least 200 mg/dL (≥11.1 mmol/L) 2 h after an oral glucose tolerance test for cases diagnosed before 1998; for cases diagnosed in 1998 and later, the fasting plasma glucose threshold was 126 mg/dL (≥7 mmol/L).
Thorand, 2006 [6]	MONICA/KORA	White	51.8 ♂/ 51.7 ♀	56.1 ♂/ 56.2 ♀	27.1 ♂/ 26.4 ♀	29.7 ♂/ 30.9 ♀	43.9 ♂/ 35.9 ♀	65.8 ♂/ 69.4 ♀	29.4 ♂/ 19.2 ♀	35.1 ♂/ 15.3 ♀	2,244	532	12	Physician diagnosis of diabetes, on participants who self-reported diabetes
Song, 2007 [27]	Women’s Health Initiative Observational Study	W: 51, B: 29, H: 12, A: 8	W:64, B:61, H:60, A:64	W:64, B:61, H:60, A:64	W:26, B:30, H:28, A:24	W:33, B: 34, H:31, A:27	W:6, B:11, H:3, A:4	W: 7, B:11, H:8, A:3	NA	NA	3,782	1584	5.5	Self-report of first-time use of hypoglycaemic medication (oral hypoglycaemic agents or insulin)
Thorand, 2007 [24]	MONICA/KORA	White	51	56	27	30	40	67	23	27	1,204	224	8	Self-report of diabetes diagnosed by a physician or intake of antidiabetic medication was validated by physician contact or medical chart review
Stranges, 2008 [28]	Western New York Study	W: 90	60	58	29	32	35	56	11	20	219	61	5.9	Fasting glucose >125mg/dL or antidiabetic medication intake
Bertoni, 2010 [57]	MESA	W: 42, C:12, B: 26, H: 21	W: 62,C: 61,AA: 62, H: 6 (among all, cases and non-cases)		W: 28, C:24, AA:30, H:29 (among all, cases and non-cases)		W: 37, C: 34, AA: 55, H:37 (among all, cases and non-cases)		W: 11, C: 5 AA: 18, H: 14 (among all, cases and non-cases)		5,571	410	4.7	Use of diabetes drugs or glucose ≥ 7 mmol/L
Dallmeier, 2012 [56]	Framingham Heart Study	W: 93, AA: 3, H: 2, A: 2	59	62	28	32	37	68	12	11	2,638	162	6.6	Fasting glucose ≥126 mg/dL or the use of insulin or oral hypoglycaemic medication
Julia, 2014 [29]	SU. VI. MAX 2	French	51	52	24.38	28.24	NA	NA	NA	NA	1,263	82	13	Fasting glucose ≥126 mg/dL or use of antidiabetic medication
Odegaard, 2016 [30]	CARDIA	B: 44	P-Selectin quartiles (1^st^ to 4^th)^: 40, 40, 40, 40	E-Selectin quartiles (1^st^ to 4^th)^: 40, 40, 40, 40	P-Selectin quartiles (1^st^ to 4^th)^: 27, 28, 29, 29	E-Selectin quartiles (1^st^ to 4^th)^: 26, 27, 29, 31	NA	NA	P-Selectin quartiles (1^st^ to 4^th)^: 12, 17, 18, 28	E-Selectin quartiles (1^st^ to 4^th)^: 11, 17, 21, 27	2,339	222 for E-Selectin and 220 for P-Selectin	10	Use of diabetes medication, fasting blood glucose ≥7 mmol/L (126 mg/dL), 2 h post-challenge glucose ≥11.1 mmol/L (200 mg/dL), or a HbA1c ≥ 6.5% (48 mmol/mol)
Pankow, 2016 [4]	MESA	W: 38, B: 28, H: 23, A: 11	P-Selectin quartiles (1^st^ to 4^th^): 61, 63, 63, 62	E-Selectin quartiles (1^st^ to 4^th^): 59, 59, 59, 58	P-Selectin quartiles (1^st^ to 4^th^): 27, 27, 27, 28	E-Selectin quartiles (1^st^ to 4^th^): 61, 63. 63. 62	NA	NA	NA	NA	E-Selectin: 826; P-Selectin: 1894	E-Selectin: 107; P-Selectin: 184	10	Use of insulin or oral diabetes medication or fasting glucose ≥ 7 mmol/L (126 mg/dL).
De Simone, 2017 [58]	Strong Heart Study	American Indians	44	47	30	35	NA	NA	NA	NA	2,887	297	4	Fasting glucose ≥126 mg/dL or the use of antidiabetic medication
Pletsch-Borba, 2019	EPIC–Heidelberg	White	50 (median)	54 (median)	25	29	34	58	23	26	2,224	163	16 (median)	Physician’s diagnosis and medical records review

Symbols ♂ represent men and ♀ women. Abbreviations: BMI body mass index, ARIC: Atherosclerosis Risk in Communities, MONIKA/KORA Multinational monitoring of trends and determinants in cardiovascular disease/”Kooperative Gesundheitsforschung in der Region Augsburg”, MESA: Multi-Ethnic Study of Atherosclerosis, SU. VI. MAX 2: Supplémentation en Vitamines et Minéraux AntioXydants, CARDIA: Coronary artery risk development in young adults, EPIC European Prospective Investigation into Cancer and Nutrition, NA not available, W white, AA African American, B Black, C Chinese, H Hispanic, WHO: world health organization. WHO guidelines incident type 2 diabetes: fasting plasma glucose ≥7 mmol/L or 2 h plasma glucose ≥11.1 mmol/L.

**Table 4 jcm-08-02075-t004:** Relationship between plasma markers of endothelial dysfunction and type 2 diabetes risk.

Author, Year	Risk Marker	Reduced Model for Type 2 Diabetes Risk (95% CI)	Multivariable-Adjusted Model for Type 2 Diabetes Risk (95% CI)	Quantiles	Reduced Model for Type 2 Diabetes Risk Continuous (per SD) (95% CI)	Multivariable-Adjusted Model for Type 2 Diabetes Risk Continuous (per SD) (95% CI)	Adjustment for Covariates (Multivariable-adj. Model)	Study Quality (max 9)	Study Design
Duncan, 1999 [52]	Fibrinogen	1.5, *p* < 0.001	1.2 (1.0, 1.5)	quartile	NA	1.07 (1.00, 1.15) *	1, 2, 4, 6, 10, 12, 23, 25	8	Cohort
Festa, 2002 [55]	Fibrinogen	NA	NA	NA	1.21 (1.01, 1.44)	1.02 (0.85, 1.24)	1, 2, 4, 8, 10	8	Cohort
Krakoff, 2003 [25]	E – Selectin	NA	NA	NA	1.12 (0.82, 1.55)	1.34 (0.91, 1.99)	1, 9, 22, 23, 24, 32	7	Nested case – control
Meigs, 2004 [5]	E-Selectin	7.50 (5.05, 11.14)	4.84 (3.06, 7.67)	Quintile	1.83 (1.37, 2.45) *	1.56 (1.12, 2.18) *	1, 6, 8, 10, 11, 25, 27, 30, 31	8	Nested case-control
Thorand, 2006 [6]	E-Selectin	3.44 (2.46, 4.83) ♂/ 2.79 (1.87, 4.17) ♀	2.79 (1.91, 4.09) ♂/ 1.72 (1.07, 2.75) ♀	Tertile	1.58 (1.24, 2.02) * ♂/1.44 (1.08, 1.93) * ♀	1.42 (1.07, 1.88) * ♂/1.18 (0.85, 1.64) * ♀	1, 3, 8, 10, 11, 12, 13, 14, 20, 25, 26	9	Case-cohort
Song, 2007 [27]	E-Selectin	5.48 (4.33, 6.94)	3.46 (2.56, 4.68)	Quartile	1.83 (1.70, 1.97)	1.58 (1.45, 1.73)	1, 4, 5, 6, 7, 8, 10, 11, 12, 25, 27	8	Nested case-control
Thorand, 2007 [24]	Thrombomodulin	NA	NA	NA	0.92 (0.78, 1.09)	0.79 (0.62, 1.00)	1, 2, 3, 8, 10, 11, 12, 13, 20, 25	8	Case-cohort
Stranges, 2008 [28]	E-Selectin	3.18 (1.32, 7.64)	2.77 (1.13, 6.79)	Tertile	1.63 (0.88, 3.00) *	1.56 (0.84, 2.91) *	1, 2, 5, 6, 8, 23, 25	8	Nested case-control
Bertoni, 2010 [57]	Fibrinogen	1.6 (1.2, 2.1) **	1.0 (0.8, 1.3)	Quartile	1.08 (0.85, 1.36) *	0.92 (0.75, 1.13) *	1, 2, 4, 6, 8, 10, 11, 12, 13, 15, 21, 28	7	Cohort
Dallmeier, 2012 [56]	P-Selectin and Fibrinogen	NA	NA	NA	P-Selectin 1.11 (0.94, 1.31) and Fibrinogen 1.24 (1.05, 1.47)	P-Selectin 0.89 (0.74, 1.07), Fibrinogen 0.91 (0.75, 1.11)	1, 2, 3, 8, 10, 13, 18, 19, 23	8	Cohort
Julia, 2014 [29]	E-Selectin	1.18 (0.67, 2.07)	1.07 (0.60, 1.93)	Tertiles	1.25 (0.96, 1.62)	1.18 (0.90, 1.55)	1, 2, 8, 19, 20, 23, 25, 38	8	Cohort
Odegaard, 2016 [30]	E-Selectin, P-Selectin	NA	E-Selectin: 2.48 (1.60, 3.85), P-Selectin: 1.48 (0.98, 2.22)	Quartiles	NA	E-Selectin 1.25 (0.90, 1.75) *, P-Selectin 1.09 (0.82, 1.46) *	1, 2, 4, 6, 10, 11, 12, 25, 33	8	Cohort
Pankow, 2016 [4]	P-Selectin, E-Selectin	E-Selectin 3.76 (1.99, 7.08), P-Selectin 1.62 (1.06, 2.48)	E-Selectin 2.49 (1.26, 4.93), P-Selectin 1.14 (0.73, 1.77)	quartiles	E-Selectin 1.24 (1.04, 1.49), P-Selectin 1.06 (0.91, 1.22)	E-Selectin 1.57 (0.95, 2.58) *, P-Selectin 1.04 (0.89, 1.21)	Cont.: 1, 2, 4, 6, 8, 9, 10, 12, 13, 15, 22, 23, 24, 26, 29. Quantiles: 1, 2, 4, 6, 8, 9, 10, 12, 13, 15, 21, 22 (also 24, 26, and 29 on E-Selectin analyses)	8	Cohort
De Simone, 2017 [58]	Fibrinogen	NA	NA	NA	NA	7.18 (1.63, 31.53)	1, 2, 4, 14, 34, 35, 36, 37	7	Cohort
Pletsch-Borba, 2019	All	see Table 2	see Table 2	see Table 2	see Table 2	see Table 2	1, 2, 8, 10, 11,12, 14, 21, 22, 26	9	Cohort

Abbreviations: CI confidence interval, NA not available, BMI indicates body mass index. Adjustment factors: ^1^ age, ^2^ sex, ^3^ Survey/cohort, ^4^ physical facility/center, ^5^ time of blood draw, ^6^ race/ethnicity, ^7^ duration of follow-up, ^8^ BMI, ^9^ waist circumference, ^10^ smoking status, ^11^ alcohol consumption, ^12^ physical activity, ^13^ SBP, ^14^ hypertension, ^15^ antihypertensive medication, ^16^ total cholesterol, ^17^ LDL, ^18^ HDL, ^19^ triglycerides, ^20^ total Cholesterol/HDL cholesterol, ^21^ education level, ^22^ HbA1c, ^23^ fasting glucose, ^24^ fasting insulin, ^25^ family history of diabetes, ^26^ CRP, ^27^ postmenopausal hormone therapy, ^28^ HOMA-IR, ^29^ IL6, ^30^ fasting status at blood draw, ^31^ diet score, ^32^ 2 h plasma glucose, ^33^ carotenoids and tocopherols, ^34^ relatedness, ^35^ impaired fasting glucose, ^36^ body fat, ^37^ visceral adiposity, ^38^ supplementation group. * Calculated from the quantiles’ analyses, as not provided by the original publication. ** Univariate model. Symbols ♂ represent men and ♀ women. Reduced model: age- and sex-adjusted, and in some cases also adjusted for survey, ethnicity, cohort, year of baseline visit, fasting status at blood draw, clinic, smoking. Multivariable-adjusted model: covariates described in column “Adjustment for covariates”.

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
