# Peer review of "Biomarkers of Vascular Injury and Type 2 Diabetes: A Prospective Study, Systematic Review and Meta-Analysis"

_jcm, 2019, doi:10.3390/jcm8122075_

Round 1
Reviewer 1 Report
Review: Pletsch-Borba et al., Biomarkers of vascular injury and type 2 diabetes: A prospective study, systematic review and meta-analysis
This study investigates the associations of 7 serum markers for vascular injury with the incidence of type 2 diabetes in a European cohort. When comparing the hazard ratios between the highest and lowest tertiles, the study showed a lower risk of type 2 diabetes in the highest tertile of ICAM3. However, in a continuous model, there was no significant relationship. E-selectin showed a non-linear association with type 2 diabetes risk. The analysis was completed by a meta-analysis for 4 of the 7 seven biomarkers. A meta-analysis for 2 of the 4 markers that found the exact same publications has been published earlier this year. Overall, the meta-analyses show a positive association of E-selectin with type 2 diabetes, no associations for P-selectin and fibrinogen. The results of the meta-analysis were accompanied by a substantial set of sensitivity analyses.
Broad comments:
The study provides a detailed description of biomarkers for vascular injury and their relation to the incidence of type 2 diabetes. It is a well-written and executed study. The introduction should be improved by some explanation about the specific choice of biomarkers (and a short description why they are markers for vascular injury). This links in with the literature review where I am missing more information on the rationale and objective. The introduction and the discussion should also refer to the fact that in the first author’s own previous cross-sectional study, three of the 7 markers have been associated with prevalent diabetes. Two of these three markers are also associated with incident diabetes. The literature review and meta-analysis are well executed and there is great set of sensitivity analyses. Unfortunately, just previously, a meta-analysis on 2 of the 4 investigated markers has been published based on the exact same papers. This is acknowledged in the manuscript. However, it needs further comparisons as to why the actual ratios in both meta-analyses are different. Also, as parts of the meta-analysis are not novel anymore, I would recommend to focus the manuscript more on the prospective analysis and either further explore the role of ICAM3 (e.g. the relation of risk factors for diabetes with ICAM3, explore the difference between cross-sectional and incidence associations), look into combining the biomarkers to enhance their predictive power or add other markers of vascular injury to increase the novelty.
Specific comments:
Line 38: The results for thrombomodulin are not very strong and the literature review only found one other study. Nonetheless this is very prominently reported in the abstract.
Line 55: The introduction is very short and only provides a reference regarding the relation of vascular injury to the biomarkers. This should be succinctly explored in more detail. Also, the authors do not provide any reasoning why they chose these 7 biomarkers and why they are not investigating others e.g. ICAM1 or VCAM1?
Line 60: I come to a different conclusion regarding the assessment of scarcity of studies. Studies that were found in the systematic review have also investigated more than one biomarker. In addition, in studies like this one that evaluate each biomarker’s predictive performance separately, the novelty argument of looking at a set of biomarkers (one by one) is not relevant to me.
Line 75: Please provide more information on the cohort selection. Why was such a comparatively small number of individuals (2,224) randomly sampled when there are over 25,000 individuals enrolled in the study? A larger study sample would improve the predictive power. This is particularly important as the cohort is relatively healthy and has few incident cases.
Line 105: The reference points to the author’s previous publication on these biomarkers and risk of MI. The correlation between biomarkers is important. With the small reported inter-correlations and the aim of the prospective study to predict diabetes incidence, I would recommend a sub-analysis on the predictive performance when these biomarkers are combined.
Line 127: Why was the minimum and maximum chosen as opposed to quantile 1 and quantile 3?
Line 147: Great to see that the authors tested to see whether the proportional hazard assumption holds.
Line 172: I am wondering why the authors do not mention that men are far more likely to develop type 2 diabetes in this cohort?
Line 178 Table 1, please provide p-values to further assess statistical significance.
Line 195 Table 2, please provide p-values for all HR results. The text should also mention that the continuous variables did not show any significant association in a fully adjusted model.
Line 260: Description for Table 3 is misleading. There are no associations presented. Please also check the number of baseline participants and incidence cases for Pankow (2016) and E-selectin. If I understand the table correctly, it currently reads that there were 82 participants and 107 cases.
Line 287: The first author’s own previous study (reference 21) has already found a cross-sectional relationship between the biomarkers and type 2 diabetes: “Individuals with diabetes at baseline had higher levels of E-Selectin, P-Selectin and ICAM3.” (page 3). This is not mentioned neither in the discussion here (only later when discussing ICAM3, but it should be mentioned here as well) when cross-sectional relationships are discussed nor in the introduction. It should be mentioned here and importantly in the introduction as well.
Line 290: Please re-word ‘lifestyle pattern’ as it is used in combination with socio-economic markers.
Line 296: Reference 45 also has an indication of a non-linear relationship in their meta-analysis. This should be further explored and the text adequately adapted.
Line 299: Please add the p-value to the statement as this is a non-significant result. It is interesting that the only other study population looking into this particular biomarker is also from southern Germany. What is the authors’ view on this? Is there any possible socio-economic link? Is there an overlap in the study cohort?
Line 305: I recommend a separate paragraph for ICAM3
Line 323: As biomarkers were not combined, I do not think it is relevant at how many biomarkers a study looks at.
Line 328: This is very important and it is unfortunate that a meta-analysis covering 2 of the 4 biomarkers has been published just recently. Because of the large overlap, this study needs to be discussed in more detailed. It is important to note that they were investigating the exact same papers in their meta-analysis. Also, there needs to be more information how the ratios differ between this meta-analysis and the recently published one.
Line 342: In my opinion the present findings for E-selectin ‘re-iterate’ rather than ‘suggest’ a relationship to type 2 diabetes as this relation has been found before.
Line 344: The wording association is too strong in my opinion. There is no association between thrombomodulin and diabetes in this study and also not in a previous study. The meta-analysis is only based on one other study.
Line 358: Who is MK?
Author Response
Response to Reviewer 1 Comments
This study investigates the associations of 7 serum markers for vascular injury with the incidence of type 2 diabetes in a European cohort. When comparing the hazard ratios between the highest and lowest tertiles, the study showed a lower risk of type 2 diabetes in the highest tertile of ICAM3. However, in a continuous model, there was no significant relationship. E-selectin showed a non-linear association with type 2 diabetes risk. The analysis was completed by a meta-analysis for 4 of the 7 seven biomarkers. A meta-analysis for 2 of the 4 markers that found the exact same publications has been published earlier this year. Overall, the meta-analyses show a positive association of E-selectin with type 2 diabetes, no associations for P-selectin and fibrinogen. The results of the meta-analysis were accompanied by a substantial set of sensitivity analyses.
Broad comments:
Point 1: The study provides a detailed description of biomarkers for vascular injury and their relation to the incidence of type 2 diabetes. It is a well-written and executed study. The introduction should be improved by some explanation about the specific choice of biomarkers (and a short description why they are markers for vascular injury). This links in with the literature review where I am missing more information on the rationale and objective. The introduction and the discussion should also refer to the fact that in the first author’s own previous cross-sectional study, three of the 7 markers have been associated with prevalent diabetes. Two of these three markers are also associated with incident diabetes. The literature review and meta-analysis are well executed and there is great set of sensitivity analyses. Unfortunately, just previously, a meta-analysis on 2 of the 4 investigated markers has been published based on the exact same papers. This is acknowledged in the manuscript. However, it needs further comparisons as to why the actual ratios in both meta-analyses are different. Also, as parts of the meta-analysis are not novel anymore, I would recommend to focus the manuscript more on the prospective analysis and either further explore the role of ICAM3 (e.g. the relation of risk factors for diabetes with ICAM3, explore the difference between cross-sectional and incidence associations), look into combining the biomarkers to enhance their predictive power or add other markers of vascular injury to increase the novelty.
Response 1: We thank the reviewer for the broad comments, which are fair and constructive. It is correct that our meta-analyses was motivated by our own cross-sectional analyses [1], and by the notion of interesting parallels with data from the MONICA/KORA Study published by Thorand et al. in 2007 [2], as also mentioned by reviewer 2. We refer to our own previous findings in more detail in the introduction now, and have also added more information on the biomarkers. While we acknowledge that we were “scooped” by the group of Qiu et al. [3] with regard to E-Selectin and P-Selectin, we do think that our analyses provide novel findings, in particular on Thrombomodulin, as we were able to add our own EPIC-Heidelberg results.
Changes in lines 56-72 and 73-83.
In fact, the results of Qiu et al. and ours are very consistent, which is a good sign in times of replication crises. That said, we think that the risk ratios presented by Qiu et al. could be different in magnitude due to a scaling phenomenon. The group showed their risks for diabetes per 1-ln μg/ml increase in biomarker levels, while our results are shown per SD log10 ng/ml. We performed a statistical simulation with fictitious data, and found that the transformation from μg to ng is probably the main reason for the different results obtained by us and our colleagues. Transforming our final meta-analysis results to per 1-log10 ng/ml and results from Qiu et al. [3] to per 1-ln ng/ml, led to only smaller differences in risk estimates. The reason for us to use the ng/ml scale was a practical one: Average biomarker levels on the μg/ml scale are very low (e.g. 0.0096 μg/ml = 9.6 ng/ml for E-Selectin among non-cases in our study). Thus, modelling increases per μg/ml may not reflect realistic differences in E-Selectin levels in the general population, and the risk ratios by Qiu et al. (e.g. 2.44 for E-Selectin) therefore appear rather high. We are aware that even in potential pooled analyses the scaling of more novel biomarkers may not be trivial, given the lack of measurement standardization. Nevertheless, we believe that the ng/ml scale we used is more appropriate for trend tests than the μg/ml scale, which we also discuss now.
Changes in lines 405-410.
We have also added ancillary analyses on the biomarkers as predictors of absolute diabetes risk, as suggested by the reviewer. Despite slight increments in the C-Statistic, we think that our results are more important from an etiological perspective than with respect to risk stratification.
Changes in lines 385-398.
The evidence on ICAM3 from human studies is rather limited, and we acknowledge that ICAM3 was not significantly associated with T2D on the linear scale. We have added more details on ICAM3 to our discussion.
Changes in lines 365-373.
Specific comments:
Point 2: Line 38: The results for thrombomodulin are not very strong and the literature review only found one other study. Nonetheless this is very prominently reported in the abstract.
Response 2: We have toned down our abstract, but do believe that the finding on thrombomodulin is interesting, as two independent studies (MONICA/KORA and ours) have shown thrombomodulin to be inversely associated with diabetes risk. Given this consistency of findings, we think that it is justified to report on the association between thrombomodulin and diabetes in the abstract, albeit in a more cautious manner. We have also added more cautious statements on thrombomodulin as potential risk factor and predictor of diabetes to the discussion and conclusion.
Changes in lines 40-42, 385-398, and 430-436.
Point 3: Line 55: The introduction is very short and only provides a reference regarding the relation of vascular injury to the biomarkers. This should be succinctly explored in more detail. Also, the authors do not provide any reasoning why they chose these 7 biomarkers and why they are not investigating others e.g. ICAM1 or VCAM1?
Response 3: We acknowledge this point, and have added more details about the biomarkers to the introduction now. The reason for us to carry out a meta-analysis on the selected seven markers was that we had data on these biomarkers from our own cohort (EPIC-Heidelberg).The measurements in EPIC-Heidelberg were initially carried out for a primary project with the aim to explore biomarkers of platelet activation in relation to different types of cancer risk [4, 5], using a case-cohort design. Motivated by cross-sectional findings on the biomarkers in relation to cardio-metabolic risk factors in a random subcohort of EPIC-Heidelberg, and considering interesting associations between a similar set of biomarkers and incident type 2 diabetes reported from the MONICA/KORA study earlier, we decided that a meta-analyses would be of interest. Our review protocol was registered on PROSPERO under CRD42019125922, and deviations from systematic review protocols have been criticized. However, given that the results by Qiu et al. on selectins are similar to ours, we would probably have produced similar findings on CAMs as well. We refer to this aspect in the discussion now.
Changes in lines 56-72, 73-82, 413-415 and 421-423.
Point 4: Line 60: I come to a different conclusion regarding the assessment of scarcity of studies. Studies that were found in the systematic review have also investigated more than one biomarker. In addition, in studies like this one that evaluate each biomarker’s predictive performance separately, the novelty argument of looking at a set of biomarkers (one by one) is not relevant to me.
Response 4: We appreciate the reviewer’s point and have rephrased our text on the motivation for the present study. The focus of our project was rather to explore the selected biomarkers as risk factors for T2D incidence, instead of developing a risk prediction model for this endpoint, which would be a next step. We believe that a meta-analysis as ours may help future prediction studies, for which biomarkers are selected in a targeted manner. We do find it interesting that biomarkers indicative of vascular injury are associated with diabetes risk over time among adults free of diabetes at baseline, as these findings suggest that vascular injury is not only a consequence of diabetes, but possibly also a risk factor beyond established risk factors.
We have now also performed sensitivity analyses testing predictive models (see below), indicating slight but statistically significant increases in the C-statistic when including E-Selectin, ICAM3 and thrombomodulin. Although these analyses may provide a rough assessment of the potential of the biomarkers for instrumental use in risk prediction, we do not think that our single cohort and meta-analysis are adequate tools to derive prediction models. We state this in the discussion and conclusion now.
Changes in lines 385-398.
Point 5: Line 75: Please provide more information on the cohort selection. Why was such a comparatively small number of individuals (2,224) randomly sampled when there are over 25,000 individuals enrolled in the study? A larger study sample would improve the predictive power. This is particularly important as the cohort is relatively healthy and has few incident cases.
Response 5: The selection of the subcohort of EPIC-Heidelberg was part of a bigger multi-biomarker project aiming to assess biomarkers of cardio-metabolic state in relation to cardiovascular disease (myocardial infarction [MI] and stroke) and cancer risk in a case-cohort design. For economic reasons, concentrations of biomarkers were not measured in samples of all participants from the EPIC-Heidelberg. By design, blood samples from a subcohort of approximately 10% of the total EPIC-Heidelberg, that was randomly selected, and from incident cases of common cancers, MI, and stroke, were used for biomarker measurements.
As diabetes was not a primary endpoint of this project, only cases observed within the randomly-selected subcohort could be used for the present analyses. While a larger sample size would have been advantageous, we do think that the case number was sufficient to detect stronger associations, particularly as the findings from EPIC-Heidelberg are consistent with those from previous studies.
We have now added more details on the selection of our sample, and discuss the lower case number as a limitation.
Changes in lines 97-102 and 411-413.
Point 6: Line 105: The reference points to the author’s previous publication on these biomarkers and risk of MI. The correlation between biomarkers is important. With the small reported inter-correlations and the aim of the prospective study to predict diabetes incidence, I would recommend a sub-analysis on the predictive performance when these biomarkers are combined.
Response 6: We have now calculated the area under the receiver operating curve (AUC) and 95% confidence intervals for the models (i) without the biomarkers of vascular injury, then (ii) including E-Selectin, ICAM3 and thrombomodulin, and finally (iii) including all seven biomarkers of vascular injury. As covariates the same as the ones used in model 2 of the risk analyses were used.
The addition of E-Selectin, ICAM3 and thrombomodulin increased significantly the AUC compared to the basal model including none of the biomarkers. The inclusion of all seven biomarkers, though, did not increase prediction of this model, compared to model I, as shown in the following table.
We have added this information to the new version of the manuscript, to the statistical methods, results and discussion.
Changes in lines 181-187, 247-252 and 385-398.
Point 7: Line 127: Why was the minimum and maximum chosen as opposed to quantile 1 and quantile 3?
Response 7: We agree that a description of percentile 25 and percentile 75 in table 1 may be more useful than minimum and maximum values, and have revised our table.
Point 8: Line 147: Great to see that the authors tested to see whether the proportional hazard assumption holds.
Response 8: Thank you.
Point 9: Line 172: I am wondering why the authors do not mention that men are far more likely to develop type 2 diabetes in this cohort?
Response 9: This information was added to the text.
Changes in line 211.
Point 10: Line 178 Table 1, please provide p-values to further assess statistical significance.
Response 10: P-values from t-tests for differences in continuous variables between T2D cases and non-cases, as well as p-values from chi-square tests for categorical variables are shown now in an extra column of table 1. Methods for these statistical analyses were added to Materials and Methods section.
Changes in lines 155-157.
Point 11: Line 195 Table 2, please provide p-values for all HR results. The text should also mention that the continuous variables did not show any significant association in a fully adjusted model.
Response 11: P-values for all HRs are described in table 2 now. A sentence on the second point was added to the manuscript.
Changes in line 230-231.
Point 12: Line 260: Description for Table 3 is misleading. There are no associations presented. Please also check the number of baseline participants and incidence cases for
Pankow (2016) and E-selectin. If I understand the table correctly, it currently reads that there were 82 participants and 107 cases.
Response 12: Table 3 description has been corrected. The typo regarding number of cases and non-cases from Pankow (2016) has been corrected: 826 participants and 107 cases (for E-Selectin).
Point 13: Line 287: The first author’s own previous study (reference 21) has already found a cross-sectional relationship between the biomarkers and type 2 diabetes: “Individuals with diabetes at baseline had higher levels of E-Selectin, P-Selectin and ICAM3.” (page 3). This is not mentioned neither in the discussion here (only later when discussing ICAM3, but it should be mentioned here as well) when cross-sectional relationships are discussed nor in the introduction. It should be mentioned here and importantly in the introduction as well.
Response 13: We thank the reviewer for pointing this out. Statements on our own previous cross-sectional analyses have been added to the introduction and the discussion.
Changes in lines 73-77 and 369-371.
Point 14: Line 290: Please re-word ‘lifestyle pattern’ as it is used in combination with socio-economic markers.
Response 14: The sentence has been rephrased, as suggested by the reviewer.
Changes in lines 341-342.
Point 15: Line 296: Reference 45 also has an indication of a non-linear relationship in their meta-analysis. This should be further explored and the text adequately adapted.
Response 15: This is correct, even though the tendency for a j-shaped association observed by Qiu et al. was not statistically significant (p=0.07). We have commented on the study Qiu et al. now in our discussion.
Changes in lines 347-353.
Point 16: Line 299: Please add the p-value to the statement as this is a non-significant result. It is interesting that the only other study population looking into this particular biomarker is also from southern Germany. What is the authors’ view on this? Is there any possible socio-economic link? Is there an overlap in the study cohort?
Response 16: The p-value for the association between Thrombomodulin and type 2 diabetes described by Thorand et al (2007) [2] was added to the manuscript, and was in fact 0.047, which we describe now. The study populations do not overlap, as they were carried out in different regions (federal states of Bavaria vs. Baden-Württemberg). However, the study populations may share similar characteristics, as suggested by the reviewer. We have added this aspect to our discussion.
Changes in lines 278 and 390-394.
Point 17: Line 323: As biomarkers were not combined, I do not think it is relevant at how many biomarkers a study looks at.
Response 17: We agree and have deleted this phrase.
Point 18: Line 328: This is very important and it is unfortunate that a meta-analysis covering 2 of the 4 biomarkers has been published just recently. Because of the large overlap, this study needs to be discussed in more detailed. It is important to note that they were investigating the exact same papers in their meta-analysis. Also, there needs to be more information how the ratios differ between this meta-analysis and the recently published one.
Response 18: We agree with the reviewer that there is an overlap between this study and the one recently published by Qiu et al., which includes meta-analyses on E-Selectin and on P-Selectin. Nevertheless, we believe that our analyses have the following additional advantages:
1. We have also included unpublished data from the prospective population-based EPIC-Heidelberg.
2. In the present analyses we also performed a systematic review on ICAM3, TPO, GPIIb/IIIa, Thrombomodulin and Fibrinogen, with meta-analyses on Thrombomodulin and Fibrinogen, for which no published meta-analysis exists.
We also discuss the differences in magnitude of risk estimates between the study Qiu et al. [3] and ours now (see reply to the reviewer’s broad comment).
Changes in lines 405-410.
Point 19: Line 342: In my opinion the present findings for E-selectin ‘re-iterate’ rather than ‘suggest’ a relationship to type 2 diabetes as this relation has been found before.
Response 19: We replaced the word “suggest” by reiterate”. This sentence has been rephrased.
Changes in lines 430-432.
Point 20: Line 344: The wording association is too strong in my opinion. There is no association between thrombomodulin and diabetes in this study and also not in a previous study. The meta-analysis is only based on one other study.
Response 20: We do think that our meta-analysis result on thrombomodulin may reflect an association. However, we agree that further studies are needed to confirm our finding, which we state now in the discussion. We are also using a more cautious wording in our abstract and conclusion now.
Changes in lines 40-42, 363-364 and 430-432
Point 21: Line 358: Who is MK?
Response 21: This was a typo, which has been now corrected.
References
1. Pletsch-Borba, L., et al., Biomarkers of vascular injury in relation to myocardial infarction risk: A population-based study. Sci Rep, 2019. 9(1): p. 3004.
2. Thorand, B., et al., Soluble thrombomodulin as a predictor of type 2 diabetes: results from the MONICA/KORA Augsburg case-cohort study, 1984-1998. Diabetologia, 2007. 50(3): p. 545-8.
3. Qiu, S., et al., Association between circulating cell adhesion molecules and risk of type 2 diabetes: A meta-analysis. Atherosclerosis, 2019. 287: p. 147-154.
4. Graf, M.E., et al., Pre-diagnostic plasma concentrations of Fibrinogen, sGPIIb/IIIa, sP-selectin, sThrombomodulin, Thrombopoietin in relation to cancer risk: Findings from a large prospective study. Int J Cancer, 2018.
5. Grafetstatter, M., et al., Plasma Fibrinogen and sP-Selectin are Associated with the Risk of Lung Cancer in a Prospective Study. Cancer Epidemiol Biomarkers Prev, 2019. 28(7): p. 1221-1227.

Reviewer 2 Report
The authors present a study in which they evaluated seven biomarkers of vascular injury to predict the risk of T2D onset. In addition, they conducted a systemic review and meta-analysis to evaluate their findings in context of other relevant studies investigating the same biomarkers for T2D onset.
I have the following major comments on the text/figure and tables, which the authors should address:
Concerning the results presented in table 2: I would like the authors to explain in more detail how these tertiles are selected. What are “sex-specific distributions”?
In addition, could they also explain which sex-specific distributions from patients categorized in tertile 2 (E-Selectin) and tertile 3 (ICAM3) associate with T2D onset?
E-Selectin was associated with higher T2D risk in the meta-analysis (HRperSD: 1.34[95%CI: 1.16, 1.54]; I2=63%) whilst a significant inverse association was observed between thrombomodulin and T2D risk in the meta-analysis (HRperSD: 0.82[95%CI 0.89, 1.07]; I2=0%.
If both E-Selectin and thrombomodulin have an association with T2D risk how do these I2 heterogeneity percentages play a role in their biomarker applicability?
The authors do not discuss this at all which they should. There is an inconsistency in the conclusion from the study (ref 13) in line 302, discussion section: it is mentioned that no statistically significant associations were found for thrombomodulin in relation to T2D. What is different with the significant association concluded in the same paper as mentioned in lines 253 – 255?
If no significant association is found (line 302), how do these results support the notion mentioned in lines 302 – 305? The authors conclude with the suggestion to conduct a multi-biomarker study.
However, are the seven biomarkers in the current study not combined to evaluate if a specific combination of these biomarkers are a better predictor of T2D onset?
It would be a significant addition to this study if the authors include such combination analyses of the evaluated biomarkers.
I also have minor comments on inconsistencies, noted while reading the manuscript, which the authors are adviced to adjust:
Table 3 Inconsistent bold text and lines used for the first row of Duncan et al. P- and E-Selectin data are incorrectly shown in the columns BMI, Age, current smokers, participants with and without diabetes at baseline for studies of Odegaard and Pankow.
Table 4. Typo observed “quartilie”, in the row of the study by Song, 2007. E-Selectin values mentioned in text (line 228) 1.58[95%CI 1.44, 1.73] do not correspond with table 1.58[95%CI 1.45, 173] (study by Song, 2007) Line 249: source of the study by Herder et al. is unknown. Please explain why this one is used instead of Thorand et al.?
In addition, supplementary figure 4 is inreadable. In the methods section, line 98, it is mentioned that a pilot study is carried out prior to the analyses presented in tables 1 and 2. Provide details on this study, eg present in a table some sample characteristics, and explain how the biological reproducibility was defined.
Author Response
Response to Reviewer 2 Comments
Point 1. The authors present a study in which they evaluated seven biomarkers of vascular injury to predict the risk of T2D onset. In addition, they conducted a systemic review and meta-analysis to evaluate their findings in context of other relevant studies investigating the same biomarkers for T2D onset.
I have the following major comments on the text/figure and tables, which the authors should address:
Concerning the results presented in table 2: I would like the authors to explain in more detail how these tertiles are selected. What are “sex-specific distributions”?
Response 1: Blood concentrations of the seven studied biomarkers were log10-transformed in order to approximate them to a normal distribution. Secondly, they were standardized (mean=0 and SD=1, by batch), in order to ease comparisons of HRs across biomarkers, and by batch to account for possible variations across batches. Lastly, ranks of the biomarkers’ concentrations were calculated for men and women separately, and categorized into tertiles. The use of sex-specific tertiles is helpful in situations where biomarker levels differ by sex, which was the case for the majority of biomarkers in our study. However, when using non-sex specific tertiles (adjusting analyses for sex), the results of our analyses on diabetes risk were similar. We motivate the use of sex-specific tertiles in the methods section now.
Changes in lines 159-161.
Point 2: In addition, could they also explain which sex-specific distributions from patients categorized in tertile 2 (E-Selectin) and tertile 3 (ICAM3) associate with T2D onset?
Response 2: We have added a supplementary table depicting sex-specific biomarker levels across tertiles (Supplementary Table 4). While our statistical analyses did not suggest interactions between sex and biomarker levels (continuous scale) in relation to diabetes risk, we must admit that the number of cases was too low for meaningful sex-stratified analyses, which we acknowledge now in the discussion.
Point 3: E-Selectin was associated with higher T2D risk in the meta-analysis (HRperSD: 1.34[95%CI: 1.16, 1.54]; I2=63%) whilst a significant inverse association was observed between thrombomodulin and T2D risk in the meta-analysis (HRperSD: 0.82[95%CI 0.89, 1.07]; I2=0%). If both E-Selectin and thrombomodulin have an association with T2D risk how do these I2heterogeneity percentages play a role in their biomarker applicability? The authors do not discuss this at all which they should.
Response 3: Our prior assumption was that the associations between these biomarkers and T2D risk may differ across different populations, which is the reason why we decided to perform a random- instead of fixed-effects meta-analysis.
Regarding the meta-analysis on E-Selectin, not only the number of included studies was higher, but also the source populations from these studies may show stronger differences. For example, Song et al. [1] reported the highest HR for this association, whose population includes a multi-ethnical cohort, in contrast to the EPIC-Heidelberg or the MONIKA/KORA study, for example. As suggested by the reviewer, the observed heterogeneity may speak against the use of E-Selectin for diabetes prediction, which we state now.
The meta-analysis on thrombomodulin showed a HR of 0.82 [95%CI 0.71, 0.95], with an I²=0%. The low I² in this case is probably due to the fact that only two studies were included, and both include populations from southern Germany. There is no overlap between the two cohorts, but similarities with regard to lifestyle, genetic background, socio-economic status etc. may explain that results from both studies on thrombomodulin were similar, which we discuss now.
As stated in the discussion, the majority of the populations included in the meta-analysis consisted of white persons, and therefore, more studies performed among persons from other ethnicities addressing this research question are needed. In addition, we think that multi-biomarker studies are needed to derive prediction algorithms, and our analyses can only give a first impression of the usefulness of the tested markers. We have added this aspect to the discussion now.
Changes in lines 354-364 and 385-398.
Point 4: There is an inconsistency in the conclusion from the study (ref 13) in line 302, discussion section: it is mentioned that no statistically significant associations were found for thrombomodulin in relation to T2D. What is different with the significant association concluded in the same paper as mentioned in lines 253 – 255? If no significant association is found (line 302), how do these results support the notion mentioned in lines 302 – 305? The authors conclude with the suggestion to conduct a multi-biomarker study.
Response 4: We apologize for this inconsistency, which was due to “borderline significant” statistical differences in our study and the one by Thorand et al. Thorand et al. [2] had actually reported an inverse association between thrombomodulin and T2D risk (0.79[95%CI 0.62,1.00] per SD increase in concentration, p=0.047). We have now rephrased this paragraph in the discussion.
Changes in lines 278, 280, 281 and 356-358.
Point 5: However, are the seven biomarkers in the current study not combined to evaluate if a specific combination of these biomarkers are a better predictor of T2D onset? It would be a significant addition to this study if the authors include such combination analyses of the evaluated biomarkers.
Response 5: We have now calculated the area under the receiver operating curve (AUC) and 95% confidence intervals for the models (i) without the biomarkers of vascular injury, then (ii) including E-Selectin, ICAM3 and thrombomodulin, and finally (iii) including all seven biomarkers of vascular injury. As covariates the same as the ones used in model 2 of the risk analyses were used.
The addition of E-Selectin, ICAM3 and thrombomodulin increased significantly the AUC compared to the basal model including none of the biomarkers. The inclusion of all seven biomarkers, though, did not increase prediction of this
We have added this information to the new version of the manuscript, to the statistical methods, results and discussion.
Changes in lines 181-187, 247-252 and 385-398.
Point 6: I also have minor comments on inconsistencies, noted while reading the manuscript, which the authors are adviced to adjust:
Table 3 Inconsistent bold text and lines used for the first row of Duncan et al. P- and E-Selectin data are incorrectly shown in the columns BMI, Age, current smokers, participants with and without diabetes at baseline for studies of Odegaard and Pankow.
Response 6: We have adjusted that in the new version of the manuscript.
Point 7: Table 4. Typo observed “quartilie”, in the row of the study by Song, 2007. E-Selectin values mentioned in text (line 228) 1.58[95%CI 1.44, 1.73] do not correspond with table 1.58[95%CI 1.45, 173] (study by Song, 2007) Line 249: source of the study by Herder et al. is unknown. Please explain why this one is used instead of Thorand et al.?
Response 7: We have now corrected the typos. There is a statement in Results, Systematic Review and meta-analysis section, where we explain the criteria used in case of multiple publications from the same study population.
Lines 263-264.
Point 8: In addition, supplementary figure 4 is inreadable.
Response 8: Supplementary Figure 4’s description was rephrased for better clarity. We also added legends to all supplementary forest plots.
Point 9: In the methods section, line 98, it is mentioned that a pilot study is carried out prior to the analyses presented in tables 1 and 2. Provide details on this study, eg present in a table some sample characteristics, and explain how the biological reproducibility was defined.
Response 9: The mentioned pilot study was published as an original article [3] and provides detailed information on the study population selection and characteristics, as well as on statistical methods, and results. Although the results of E-Selectin and ICAM3 were not published, data was available and results were then mentioned in our already published article on biomarkers of vascular injury and myocardial infarction risk [4]. In the following table, which was added to the supplementary material (Supplementary Table 4), this information is summarized:
*For analyses on disease risks, samples of cases and non-cases were randomly assigned to analytical batches to avoid differential misclassification. Thus, between-batch variation could be addressed by statistical batch-standardization.
**Derived from a subsample of n=78 [3, 4]
References
1. Song, Y., et al., Circulating levels of endothelial adhesion molecules and risk of diabetes in an ethnically diverse cohort of women. Diabetes, 2007. 56(7): p. 1898-1904.
2. Thorand, B., et al., Soluble thrombomodulin as a predictor of type 2 diabetes: results from the MONICA/KORA Augsburg case-cohort study, 1984-1998. Diabetologia, 2007. 50(3): p. 545-8.
3. Graf, M.E., et al., Biological reproducibility of circulating P-Selectin, Thrombopoietin, GPIIb/IIIa and Thrombomodulin over one year. Clin Biochem, 2017. 50(16-17): p. 942-946.
4. Pletsch-Borba, L., et al., Biomarkers of vascular injury in relation to myocardial infarction risk: A population-based study. Sci Rep, 2019. 9(1): p. 3004.

Round 2
Reviewer 1 Report
I want to thank the authors for addressing my comments in the manuscript and providing comprehensive answers and further analyses to all my questions in the report.